# A Hybrid Adaptive Transaction Injection Protocol and Its Optimization for Verification-Based Decentralized System †

**Saumendra Sengupta [1], Chen-Fu Chiang [1,\*], Bruno Andriamanalimanana [1], Jorge Novillo [1] and Ali Tekeoglu [2]**

1   Computer Science, State University of New York Polytechnic Institute, Utica, NY 13502, USA
2   Network and Computer Security, State University of New York Polytechnic Institute, Utica, NY 13502, USA
\*   Correspondence: chiangc@sunypoly.edu
†   This paper is an extended version of Parameterized Pulsed Transaction Injection Computation Model and Performance Optimizer for IOTA-Tango published in 3PGCIC-2018.

**Abstract:** Latency is a critical issue that impacts the performance of decentralized systems. Recently we designed various protocols to regulate the injection rate of unverified transactions into the system to improve system performance. Each of the protocols is designed to address issues related to some particular network traffic syndrome. In this work, we first provide the review of our prior protocols. We then provide a hybrid scheme that combines our transaction injection protocols and provides an optimal linear combination of the protocols based on the syndromes in the network. The goal is to speed up the verification process of systems that rely on only one single basic protocol. The underlying basic protocols are Periodic Injection of Transaction via Evaluation Corridor (PITEC), Probabilistic Injection of Transactions (PIT), and Adaptive Semi-synchronous Transaction Injection (ASTI).

**Keywords:** decentralized; optimization; synchronous; blockchain

---

## 1. Introduction

Distributed systems provide convenience for dealing with online activities. With some enhanced techniques, such as cryptography, they provide secure and scalable architectures with a wide applicability. One of the most prominent examples is the crypto-currency system. In recent years, distributed crypto-currency systems have provided an architecture that offers transparency and enables a communication ecosystem that generates billions of transactions. For instance, Internet of Things (IoT) Tangle [1] is one architecture that uses a distributed Directed Acyclic Graph (DAG) structure. However, distributed systems might suffer from the lack of synchronization in databases. To overcome this disadvantage, a semi-synchronous architecture IOTA-Tango was introduced in [2] based on the basic architecture of IOTA-Tangle.

Synchronicity is achieved when the transactions for validation are picked by the validators without ever being idle. The goal is to minimize the average waiting time for transactions without ignoring any transactions. It is important that the waiting time for an average transaction is minimized and no transaction is ignored. To ensure the system is synchronous with respect to the new transactions, an assignment mechanism was designed for scheduling those transactions to the verifiers with these following characteristics

- transactions are seen across the underlying ledger system;
- first come first serve mode is used to release the transactions to the verifiers before they are sent to the controller;

- the placement order of transactions in Tango is decided by the controllers.

Adjacent steps (processes) must be mutually synchronous.

In this paper, a brief review of our prior work on efficient injection protocols is provided. The Periodic Injection of Transaction via Evaluation Corridor (PITEC) was first introduced in [2]. The Probabilistic Injection of Transactions (PIT) was introduced in [3]. In this work, Adaptive Semi-synchronous Transaction Injection (ASTI) and Hybrid Adaptive Injection (HAI) protocols are further introduced. The rest of the paper is organized as follows. The reviews on PITEC and PIT are provided at Sections 3 and 4, respectively. A discussion on the ASTI protocol is at Section 5. For ASTI, the injection amount and the pulse period are used as the parameters for regulating the injection process. The hybrid protocol is introduced in Section 6 and finally, our conclusions are presented in Section 7.

## 2. Structure of Tango

Tango is a distributed ledger architecture that is similar to the Iota-tangle design as articulated in [1]. Transactions are released to a system for validation and subsequent affixation to Tango. The lifetime of a typical transaction has three types (1) *unevaluated* (2) *evaluated*, and (3) *committed*. Upon arriving to the system, the transactions are immediately visible to all the evaluators. In [2] the *decentralized semi-synchronous pulse diffusion* (DSPD) protocol was proposed to make the system more semi-synchronous. The DSPD protocol aims at providing a more synchronous arrival of transactions via scheduled injection. Interested readers can refer to that work.

While the transactions are inside the system awaiting verification, it is necessary to lower the latency of the system. PITEC, PIT and ASTI protocols help the controllers regulate injection of unevaluated transactions to the verifiers. The scheduling, both in terms of quantity and time, of the injections by the controllers determines how the system would reach stability (equilibrium). The injection/scheduling problems are quite similar to logistic problems in supply chain research [4,5].

## 3. PITEC Protocol

From the perspective of verification process, the major participants are controllers, Verifiers, and standby-ers. The depletion rate $D$ is assumed to be *constant* in this protocol. The controllers regulate the traffic and are selected randomly among the participants. The user's role changes as time evolves. The controllers regulate the traffic via *periodically* injecting unverified transactions to the verifiers. Associated with pulse period, $T$, there are a fixed cost $A$ and a variable cost $v$ per cycle. The fixed cost $A$ is irrespective of the volume of injected transactions. The variable cost $v$ is associated with exposing the transactions to any vulnerability. $v$ is proportional to the time a transaction remains as a tentative before it joins the DAG as a leaf node. Let $Q^*$ be the cost-optimum volume of the total unverified transactions available at the beginning of each cycle. For a theoretical simulation purpose, the *exponential smoothing* approach is used. Exponential smoothing is a well-known technique for analysis of time-series data. It predicts the number of unverified transactions released into the system. Transactions that failed the verification or had to be returned represent at least an opportunity cost to the system. They must be reflected in the model.

The total cost, $C$, per unit time $T$ is

$$C = \frac{A}{T} + \frac{Q \times v}{2},\qquad(1)$$

where $\frac{1}{2}Q \times T$ is the average volume of transactions vulnerable during the cycle. Since the total expected vulnerable time for $Q$ transactions is $(1/2)(QT)$, the expected total cost associated with vulnerability is $(1/2)(QvT)$. Therefore, the total cost is

$$C_{total} = A + \frac{1}{2}Q \times v \times T.\qquad(2)$$

During the cycle time $T$ and a fixed consumption rate $D$, the number of consumed transactions is $Q = D \times T$. Therefore, the cost per unit time is

$$C = \frac{A \times D}{Q} + \frac{1}{2}Q \times v. \tag{3}$$

This leads to

$$Q^* = \sqrt{\frac{2A \times D}{v}}, \quad T = \sqrt{\frac{2A}{v \times D}}. \tag{4}$$

## 4. PIT Protocol

For PIT [3], the expected depletion rate $D$ is *not constant*. The rationale behind PIT is to resolve the shortage issue where PITEC might encounter. The core tasks to embed in the protocol design would be (1) the time to replenish and (2) the quantity of transactions to be replenished. Equilibrium finding within a cycle provides the solutions to the two tasks. To use the verifier resources, one must calibrate the optimum to replenish for the shortage scenarios. In PIT protocol, the randomization is invoked with a basic assumption. The basic assumption is that the expected average transaction validation rate is a stable constant, D, even though the underlying micro-level behavior is random as illustrated in Figure 1.

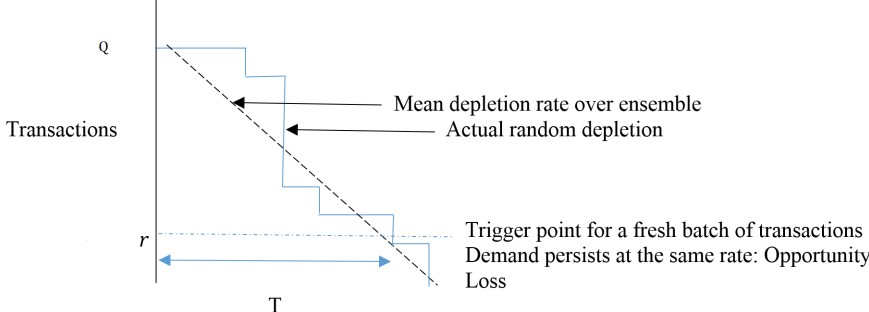

**Figure 1.** Q on the Y axis is the number of transactions to be verified while X axis is the time. T represents the time for a cycle. *D* is the mean depletion rate (the dashed line) of random rates. When less than *r* transactions left unconsumed, the trigger is activated.

When a transaction level $r$ is reached, this triggers the optimum level, $Q^*$, of fresh transaction release. Let lead time, $L$, be the time it would take from the time level $r$ is reached to the time fresh transactions are released. For this model, we have couple variable factors to consider. First $L$ is random and is predicted by how fast controller could provide the supply. Second, the depletion rates at the evaluators are random. To simplify the analysis, PIT protocol assumes that each cycle is of fixed time length $T$. A typical layout of such a model may be found in [6] for inventory control system. The useful variables are summarized in Table 1.

With a fixed cycle time $T$, this immediately resolves the first task about when to replenish into the system. For the second task, what is left is how to predict the optimum pulsed batch size $Q^*$. In this section, a sketch is provided. Interested readers should refer to the original article [3]. Let $p(y)$ be a probability mass function that describes the random depletion of transactions. With probability $p(y)$, $y$ is the number of transactions verified during the lead time $T$. The time movement profile of transactions and its random depletion rate is shown in Figure 2.

**Table 1.** Variables.

| | **Variable Description** |
|---|---|
| $A$ | Fixed cost per cycle |
| $C$ | Cost per cycle per unit time |
| $F$ | Cost per cycle |
| $L$ | Lead time: time elapsed between reorder and replenishment |
| $M_L$ | Average volume of verified transactions during L |
| $Q^*$ | Optimum batch size |
| $Q$ | Optimum batch size level |
| $S_L$ | Expected number of shortage transactions |
| $T$ | Time length of a cycle |
| $T_r$ | Time elapsed between replenishment and reorder |
| $c_s$ | Unit shortage cost during lead time |
| $r$ | Trigger threshold |
| $v$ | Cost of carrying released but unverified transactions to next cycle per transaction per unit time |

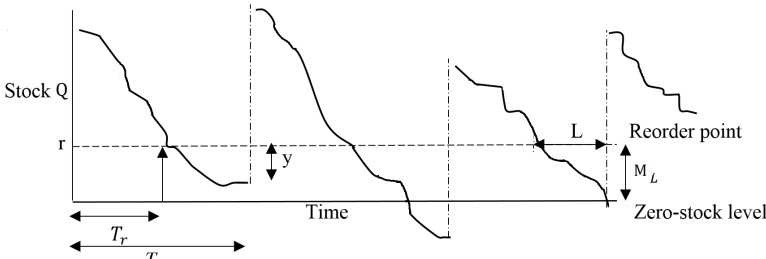

**Figure 2.** Q on the Y axis is the number of transactions to be verified while X axis is the time. T represents the time for a cycle. $T_r$ is the trigger time when $r$ transactions are left unconsumed. The second plot shows a shortage situation. $M_L$ is the average consumption during $L$ where $T_r + L = T$.

During the lead time $L$, let the average number of verified transactions be

$$M_L = \sum_{y=0}^{\infty} p(y)y,$$ (5)

and let the average volume of the unverified transactions be

$$\frac{1}{2}(r + \sum_{y=0}^{r} (r - y)p(y)).$$ (6)

The expected number of transactions available to the verifiers after a replenishment until the next reorder trigger is

$$\frac{1}{2}(\underbrace{(r + Q - M_L)}_{\text{transactions after replenishment}} + \underbrace{r}_{\text{transactions at reorder}}) = \frac{(2r + Q - M_L)}{2}$$ (7)

Thus, between replenishment and reorder, the number of transactions consumed is

$$T_r \times D = (Q + r - M_L) - r = Q - M_L,$$ (8)

and it is known that

$$(T - T_r) \times D = M_L. \tag{9}$$

During the lead time $L$, the expected number of deficient transactions is

$$\sum_{y=r+1}^{\infty} p(y)(y - r) = S_L. \tag{10}$$

while

$$\sum_{y=0}^{r} p(y)(r - y) = (\sum_{y=0}^{\infty} - \sum_{y=r+1}^{\infty}) p(y)(r - y)$$

$$= r - M_L + S_L, \tag{11}$$

Let $F$ be the transaction cost per cycle (with fixed and variable costs). Assume the unit shortage cost during $L$ is $c_s$. One can obtain

$$F = A + \frac{v}{2}[(2r - M_L + Q)T_r + (2r - M_L + S_L)(T - T_r)] + c_s \times S_L \tag{12}$$

from Equations (6), (7) and (11). The total cost per cycle per unit time is

$$C = \frac{A \times D + c_s \times S_L \times D}{Q} + \frac{v \times M_L}{2Q}(2r - M_L + S_L) + \frac{v}{2Q}(2r + Q - M_L)(Q - M_L) \tag{13}$$

because $C = F/T$, $Q = DT$ and Equations (8) and (9). One can further rewrite Equation (13) as

$$C = \frac{A \times D + c_s \times S_L \times D + \frac{v}{2} \times M_L \times S_L}{Q} + \frac{v \times Q}{2} + v(r - M_L) \tag{14}$$

where $Q^*$ can be determined by solving $\frac{dC}{dQ}$ that

$$Q^* = \sqrt{\frac{2A \times D}{v}(1 + \frac{c_s \times S_L}{A} + (\frac{v}{2A \times D})M_L \times S_L)}. \tag{15}$$

## 5. ASTI Protocol

The ASTI protocol [7] aims at predicting the injection volume based on past consumption records. It applies the technique smoothing factor, which is similar to reinforcement learning. One of the strategies ASTI uses is classifying transactions into various categories. For simplicity, let us assume it is transactions of $\theta$ type being processed while its corresponding batch size is $q_\theta^*$ and its periodicity is $T_\theta^*$ under the PITEC setting. Each class of transactions of type $\theta$ has its optimum performance that requires periodic injection of $q_\theta^*$ unverified transactions in every cycle.

Suppose the last cycle has a time lag of $T_{last}$ while $q_{last}$ transactions were injected in the last cycle. If the performance of the network remains the same as last cycle, one would need to adjust the injection volume that

$$q_{(next,\theta)} = q_{(last,\theta)} \frac{T_\theta^*}{T_{(last,\theta)}}. \tag{16}$$

However, this is naive because one cannot simply predict the performance of the network based on only previous cycle that Equation (16) proposes. To consider effects from more earlier cycles for the performance prediction, one can do

$$\hat{q}_{(next,\theta)} = \gamma q_{(last,\theta)} \frac{T_\theta^*}{T_{(last,\theta)}} + (1 - \gamma)\hat{q}_{(last,\theta)} \tag{17}$$

where $0 < \gamma < 1$ with initial expectation value

$$\hat{q}_{(last,\theta)} = q_\theta^*, \quad T_{(last,\theta)} = T_\theta^* \tag{18}$$

One can visualized the process as follows (Figure 3):

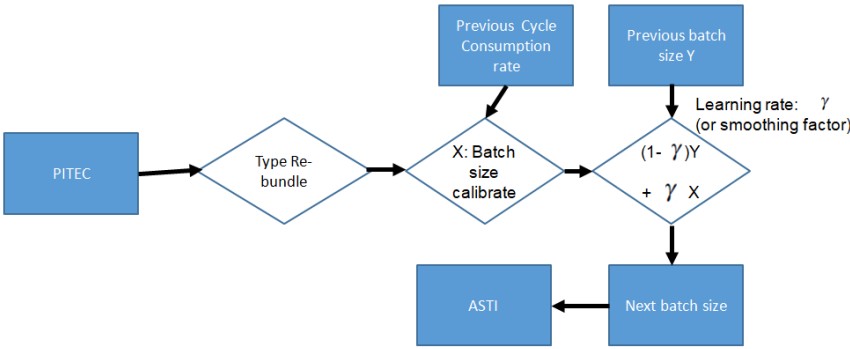

**Figure 3.** The ASTI process: smoothing factor based.

ASTI protocol can be further extended to multiple categories of transactions, based on the nature of the transactions. Some transactions might have more importance than others. This could be extended further as future work for another optimization scheme based on the importance of the types. Furthermore, initially when the unverified transactions are entered into the system, the representation of the system can be treated as a temporal graph. One can further improve the temporal graph by the Pregel-based approach in [8].

## 6. HAI Protocol

The previous three protocols are all under the same assumption of the system: a fixed cycle time $T$. In this section, it is assumed the system still uses a fixed cycle time $T$. If not, the system uses dynamic injection time where injection is triggered in the system upon the completion of the verification of the very last unverified transaction of the current batch. The system immediately injects another batch of a fixed number of unverified transactions into the system to be verified.

Under the fixed cycle time $T$ constraint, let the optimal injection amount, based on its own scenario described in the protocol, be $Q_1$ for PITEC, $Q_2$ for PIT protocol and $Q_3$ for ASTI protocol. Denote the hybrid adaptive injection protocol (HAI). HAI injects $Q_{123}$ transactions into the system at the beginning of each cycle while the cycle time is $T$. The dynamics of PIT and ASTI are driven by the change of consumption rate $D$. The change of consumption rate can be identified with the existence of $S_L$ in PIT while it is via $T_\theta^* / T_{(last,\theta)}$ in ASTI. For simplicity, let us simply use $T/T_{last}$ for ASTI at the moment without specifying the transaction type. PIT immediately reacts to the unverified transaction shortage problem as it collapses to the PITEC instance when $S_L \to 0$. On the other hand, ASTI responds to both shortage problems and overflow problems (unfinished unverified transactions at the end of the cycle) as the volume of injection bundle is regulated by $\gamma$ and $T/T_{last}$.

Considering the relation between $Q_1$ and $Q_2$, one can interpolate between them via the ratio $S_L/M_L$. The ratio $S_L/M_L$ indicates the severity of the shortage issue in PIT as indicated in Section 4. Similar to the smoothing factor technique, here the ratio works as a good smoothing factor that serves the purpose. Let the interpolated quantity be

$$Q_{12} = (1 - \frac{S_L}{M_L})Q_1 + \frac{S_L}{M_L}Q_2. \tag{19}$$

This remedy has a quite strong reliance on the assumption that the consumption pattern in previous cycle will persist in the current cycle. One drawback is that $Q_{12}$ cannot reflect the scenario

when there is an overflow issue in the system. It is necessary to incorporate the ASTI protocol. The key task is to identify the overflow indicator. It is clear that the ratio $T/T_{last}$ can serve for that purpose. When $0 < T/T_{last} < 1$, it is an overflow issue. It is natural to inject significantly fewer transactions in order not to paralyze the system. For that reason, the PIT and the PITEC protocols should be avoided. Define the binary variable

$$\delta = \begin{cases} 0 & \text{if } 0 < T/T_{last} < 1, \\ 1 & \text{if } T/T_{last} \geq 1. \end{cases} \tag{20}$$

The HAI protocol will inject

$$Q_{123} = (\delta)((1 - \frac{S_L}{M_L})Q_1 + \frac{S_L}{M_L}Q_2) + (1 - \delta)Q_3 \tag{21}$$

transactions to the system to be verified.

## 7. Conclusions

In this work, a brief introduction on the Tango system is given and reviews on PITEC, PIT, and ASTI protocols are provided. These protocols are designed for a fixed cycle time $T$ of the system for scenarios with assumptions on the consumption speed. The three protocols vary in the assumption of the network ecosystem. PITEC assumes a constant consumption rate, while PIT takes a probabilistic approach while ASTI applies the smoothing factor technique (simple reinforcement learning) for prediction. Each protocol has its own assumptions regarding the traffic behavior of the network. Without assuming the condition of the consumption rate in the network, an HAI protocol is introduced. HAI is a linear combination of the three aforementioned protocols. Future work will involve the implementation of the hybrid model for obtaining the empirical data and fine tuning for finding a good smoothing factor $\gamma$.

**Author Contributions:** Conceptualization, S.S. and C.-F.C.; methodology, S.S. and C.-F.C.; validation, S.S. and C.-F.C.; formal analysis, S.S. and C.-F.C.; investigation, B.A., C.-F.C., S.S., J.N. and A.T.; writing—original draft preparation, C.-F.C.; writing—review and editing, S.S. and C.-F.C.; supervision, S.S.

**Funding:** This research received no external funding.

**Acknowledgments:** The authors gratefully acknowledge support from the State University of New York Polytechnic Institute.

**Conflicts of Interest:** The authors declare no conflict of interest.

## Abbreviations

The following abbreviations are used in this manuscript:

| | |
|---|---|
| PITEC | Periodic Injection of Transactions via Evaluation Corridor |
| PIT | Probabilistic Injection of Transactions |
| ASTI | Adaptive Sem-synchronous Transaction Injection |
| HAI | Hybrid Adaptive Injection |

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
