# Peer review of "A Hybrid Adaptive Transaction Injection Protocol and Its Optimization for Verification-Based Decentralized System†"

_futureinternet, doi:10.3390/fi11080167_

Reviewer 1 Report

This paper claims to provide a follow-up work of the authors on a distributed ledger architecture to support better transactions. However, this seems to be a review paper for prior work of the authors.

For an archival paper that is follow-up, or that shows all of the pieces of work, the paper should have circa 30% novelty, and that novelty should include: experimentation.

AS it stands, the paper is still not ready for archival.

Some detected typos and technical aspects that need to be checked.

Typos:

- line 13, distributed systems is plural --> distributed systems proviDE

- line 14, scalable architectures for many uses.--> with a wide applicability?

- IOT-Tangle -> define acronyms the first time they are used

- directed acyclic graph (DAG) --> use major lettering

- avoid writing "we"

To assure the system --> to ensure that the system...

SOME Technical aspects to be checked/corrected:

- line 19, asynchronous communication is actually a must for a decentralized Internet. See the case of IoT, which is mentioned in the paper. Lack of sync in databases or state is something else, and possibly what the authors are mentioning.

- line 97, given that 12 Q × T is the average volume of transactions vulnerable during the cycle. -- where did the authors get the 1/2 Q value? Please explain and/or add references. Otherwise this is just an hypothesis that has not been proven.

- Eq 2 needs to be better explained. Q is a VOLUME for transactions. While AxD is actually a cost, not a number of transactions. The formula does not seem to be Ok,.

- On eq 3, what is v? The formula needs to be better explained *closer to the formula* (it is on Table 1, but this format does not help the reader in understanding the rationale of each equation).

- Figure 1 is not readable.

- Figure 2 is not readable

Author Response

For an archival paper that is follow-up, or that shows all of the pieces of work,
the paper should have circa 30% novelty, and that novelty should include: experimentation.
AS it stands, the paper is still not ready for archival.

Ans:
This work provides review on existing protocols and we introduced the new HAI protocol (section 6) that has not appeared in any other venues. Regarding the empirical data from the physical implementation, we hope this will be our near future work to attack. We do understand theory should be tested by experiments.

Some detected typos and technical aspects that need to be checked.
Typos:
- line 13, distributed systems is plural --> distributed systems proviDE
- line 14, scalable architectures for many uses.--> with a wide applicability?
- IOT-Tangle -> define acronyms the first time they are used
- directed acyclic graph (DAG) --> use major lettering
- avoid writing "we"
To assure the system --> to ensure that the system...

Ans:
The typos have been corrected as suggested by the reviewer. The use of "we" has been avoided.

SOME Technical aspects to be checked/corrected:
- line 19, asynchronous communication is actually a must for a decentralized Internet.
See the case of IoT, which is mentioned in the paper. Lack of sync in databases or state is
something else, and possibly what the authors are mentioning.

Ans:
As suggested by the reviewer, we changed from asynchronous communication to lack of synchronization in the database. We also cleaned up the original section 1 and section 2 to make sure the article flow is decent without clumsy sentences.

- line 97, given that 12 Q × T is the average volume of transactions vulnerable
during the cycle. -- where did the authors get the 1/2 Q value? Please explain and/or
add references. Otherwise this is just an hypothesis that has not been proven.

Ans: the unit is transaction-time, so, for a Q quantity of transactions that is consumed
in time T, the average (expected) transaction-time is 1/2 (QT)[that is the triangle area] since at t=0, quantity is Q and when t=T, the quantity is 0. One can interpret this as average transaction-time. On the other hand, one can also read it as the following:  the transactions to be verified are t_1, t_2, ..., t_Q (let say this is also the order they got consumed). That means t_Q has been waiting for almost T time in the cycle before it is consumed while t_1 maybe almost 0 waiting time. Hence, the expected volume for transaction being vulnerable is 1/2 (QT). We added an extra equation to show what the total cost is then the total cost is divided by the cycle time T to show the unit time cost.

- Eq 2 needs to be better explained. Q is a VOLUME for transactions.
While AxD is actually a cost, not a number of transactions. The formula does not seem to be Ok,.

Ans: The definition of A is constant cost (or let say, simply name is as set up cost for instance).
We know the consumption rate is D (that is Q transactions got consumed in T time; so D = Q/T).
So, we know that 1/T = D/Q. By use of direct substitution for equation 1 to replace 1/T, we immediately
obtain equation 3 (in older version it is equation 2) from equation 1. To make it more apparent to readers,
we added equation 2 in the revised version showing the total cost (prior to dividing by T)

- On eq 3, what is v? The formula needs to be better explained *closer to the formula*
(it is on Table 1, but this format does not help the reader in understanding the rationale
of each equation).

Ans:
We moved the definition of v closer to equation 3 and provided a detailed explanation between
equation 1 and 2.

- Figure 1 is not readable.
Ans: More explanation is given at the figure caption to make it more readable by explaining
what x axis means and what y axis means. And the meaning of the variables given in the figure.

- Figure 2 is not readable
Ans: More explanation is given at the figure caption to make it more readable by explaining
what x axis means and what y axis means. And the meaning of the variables given in the figure.

Reviewer 2 Report

The paper is quite well written, and I have no problem following the paper. As someone in operations research, I feel the described approach is quite similar to logistic problems in supply chain research. So I would suggest the authors add some references in this part

On page 6, "

We have shown that even when the underlying verification process is probabilistic with a stable average verification rate D, the stochastic periodic injection transaction (PIT) process is basically identical to its deterministic version."

I am actually quite surprised by this conclusion, could you add the reference to this conclusion? When you say that the stochastic PIT process is identical to the deterministic version: does that mean the expected value is the same? variance? Please elaborate

Author Response

Q:
The paper is quite well written, and I have no problem following the paper. As someone in operations research, I feel the described approach is quite similar to logistic problems in supply chain research. So I would suggest the authors add some references in this part

Ans:
At the end of section 2, we cited reference 10 (Logistics and Supply Chain Management) and reference 11 (Global Logistics and Supply Chain Management).

On page 6,
"We have shown that even when the underlying verification process is probabilistic with a stable average verification rate D, the stochastic periodic injection transaction (PIT) process is basically identical to its deterministic version."

I am actually quite surprised by this conclusion, could you add the reference to this conclusion? When you say that the stochastic PIT process is identical to the deterministic version: does that mean the expected value is the same? variance?
Please elaborate

Ans:
When the average rate is D, it implies the total consumption (numbers of transactions) for that cycle is DT. The author added a sentence and specified this particular DT number is exactly the number of unverified injected transactions to be consumed at the beginning of each cycle such that the variable S_L will be 0 and the PIT becomes PITEC.